# Adaptive LDA Classifier Enhances Real-Time Control of an EEG Brain–Computer Interface for Decoding Imagined Syllables

**DOI:** 10.3390/brainsci14030196

**Published:** 2024-02-21

**Authors:** Shizhe Wu, Kinkini Bhadra, Anne-Lise Giraud, Silvia Marchesotti

**Affiliations:** 1Speech and Language Group, Department of Basic Neurosciences, Faculty of Medicine, University of Geneva, 1211 Geneva, Switzerland; shizhe.wu@unige.ch (S.W.); kinkini.bhadra@unige.ch (K.B.); anne-lise.giraud@unige.ch (A.-L.G.); 2Institut de l’Audition, Institut Pasteur, Université Paris Cité, Inserm, 75012 Paris, France

**Keywords:** brain–computer interface, adaptive LDA classifier, electroencephalography, speech imagery, syllable decoding

## Abstract

Brain-Computer Interfaces (BCIs) aim to establish a pathway between the brain and an external device without the involvement of the motor system, relying exclusively on neural signals. Such systems have the potential to provide a means of communication for patients who have lost the ability to speak due to a neurological disorder. Traditional methodologies for decoding imagined speech directly from brain signals often deploy static classifiers, that is, decoders that are computed once at the beginning of the experiment and remain unchanged throughout the BCI use. However, this approach might be inadequate to effectively handle the non-stationary nature of electroencephalography (EEG) signals and the learning that accompanies BCI use, as parameters are expected to change, and all the more in a real-time setting. To address this limitation, we developed an adaptive classifier that updates its parameters based on the incoming data in real time. We first identified optimal parameters (the update coefficient, UC) to be used in an adaptive Linear Discriminant Analysis (LDA) classifier, using a previously recorded EEG dataset, acquired while healthy participants controlled a binary BCI based on imagined syllable decoding. We subsequently tested the effectiveness of this optimization in a real-time BCI control setting. Twenty healthy participants performed two BCI control sessions based on the imagery of two syllables, using a static LDA and an adaptive LDA classifier, in randomized order. As hypothesized, the adaptive classifier led to better performances than the static one in this real-time BCI control task. Furthermore, the optimal parameters for the adaptive classifier were closely aligned in both datasets, acquired using the same syllable imagery task. These findings highlight the effectiveness and reliability of adaptive LDA classifiers for real-time imagined speech decoding. Such an improvement can shorten the training time and favor the development of multi-class BCIs, representing a clear interest for non-invasive systems notably characterized by low decoding accuracies.

## 1. Introduction

### 1.1. Brain–Computer Interfaces for Speech Decoding

Neurological disorders such as locked-in syndrome (LIS) and aphasia [1] can impair the ability to communicate with the external world while leaving cognitive functions, including consciousness, intact [2]. Harnessing these residual cognitive capabilities has the potential to restore communication, thereby enhancing patients’ overall quality of life. A promising approach is to use a speech brain–computer interface (BCI), a system able to provide the user with a means of communication by decoding neural activity related to speech intentions directly from brain recordings, without the involvement of the motor system. Recent breakthroughs have shown impressive performance in real-time decoding of attempted speech [3,4,5,6], reaching a decoding rate of 78 words per minute [6]. 

Despite its effectiveness, decoding attempted speech is precluded in patients with language disorders in which the damage is located upstream with respect to the motor representations, such as in aphasia. In these patients, a more appropriate approach would be to decode imagined (i.e., covert) rather than attempted speech. Speech imagery is defined as the silent, internal generation of speech elements without any audible output or associated physical articulation [7]. Speech imagery entails less cortical activation and different cortical representations than overt, articulated speech [8,9,10,11,12]. These characteristics make covert speech remarkably more difficult to decode from neural signals. Thus, it is not surprising that BCI based on covert rather than overt speech lags behind in terms of research breakthroughs and developments. Real-time speech synthesis with acceptable intelligibility from speech imagery remains an important goal to be achieved [13].

Despite these challenges, previous studies have delved into the complexities of covert speech decoding, using both intracranial recordings [14,15,16] and non-invasive techniques such as EEG [17,18,19,20,21], employing the imagery of different types of speech units such as vowels [22,23], syllables [24,25], words [16,26,27,28], and sentences [29]. Only two previous BCI studies based on surface EEG recordings have so far attempted to decode imagined speech in real time, with limited effectiveness [18,30]. Many more studies have targeted the decoding of imagined speech with offline analyses, with the main goal of enhancing current classifiers, but none have tested their effectiveness in improving BCI control in real time. Yet, this is a crucial aspect to consider owing to EEG non-stationarity [31], which can hamper decoding in real time.

### 1.2. Non-Stationarity of EEG Signals and Adaptive Classifiers

Significant variations in the EEG signals occur throughout the course of an experiment due to different reasons. These include the hardware employed to record brain signals through EEG (e.g., drying of the conductive gel or shifts in the position of the electrodes), physiological fluctuations in cognitive states (such as attention, motivation, and vigilance levels), and neural changes due to short-term brain plasticity resulting from practicing the task [32]. In addition, BCI experiments can be affected by yet another source of variability, namely the experimental discrepancy between the *offline training* part, in which data are acquired to calibrate the classifier without providing feedback to the user (open loop), and the *online control* part, in which the user is given a feedback in real time (closed loop). For these reasons, a classifier developed at the beginning of the experiment based on offline data might not be able to optimally decode EEG signals in real time.

To mitigate the effects of non-stationarities, the use of adaptive machine learning techniques in BCI systems has emerged as a promising solution. Compared to static classifiers that remain constant throughout the duration of the experiment, adaptive BCI systems continuously update either their decoding features or classifier parameters based on new incoming signals during real-time control [33,34]. Unlike their static counterparts, these approaches enable a co-adaptive calibration between the user and the machine during online control, enhancing both BCI controllability [35,36] and signal decodability. The effectiveness of these decoding methods has been established in previous motor-imagery BCI studies [36,37,38], which highlighted that decoding improvements with an adaptive classifier not only help users with poor BCI control but can also speed up the learning curve in good BCI users [35,36,39]. 

In terms of feature extraction, methodologies such as adaptive common spatial patterns [40,41], supervised feature extraction [42], adaptive autoregressive parameters [43], and adaptive wavelet packet transform [44] have shown their effectiveness in improving the discriminability of decoding features. The other approach consists of using adaptive classifiers that update their parameters (e.g., the weights assigned to each feature in the linear discriminant hyperplane) whenever new data are available. These methods have proven to consistently perform well with non-stationary EEG signals [33,45]. Different types of linear classifiers can be translated into their adaptive counterparts, including linear discriminant analysis (LDA) [32,46,47], support vector machines [48], and Bayesian classifiers [49]. 

Given the success of adaptive classifiers in motor imagery BCIs, similar benefits can be expected in speech imagery BCIs [17], where decoding is notably difficult. 

Thus, we conducted a BCI control study to test the hypothesis that using an adaptive rather than a static LDA classifier would be effective in enhancing the control of a binary speech BCI based on the imagery of two syllables from EEG signals. We first fine-tuned the parameters of the adaptive LDA classifier using open-loop simulations performed using a prerecorded dataset. Subsequently, we tested for performance improvement in real time by comparing BCI control using the static and the adaptive LDA with these optimized parameters in a group of twenty healthy volunteers, each of whom underwent two BCI control sessions, using each classifier on separate days. Last, we explored an alternative approach for the optimization of adaptive classifiers based on individualized tuning of the classifier parameters. 

This research addresses the limited effectiveness of current approaches to decode imagined speech in real time, an important challenge that deserves prompt solutions.

## 2. Materials and Methods

### 2.1. Participants

We first used EEG data from a previous study (Study 1) conducted in our laboratory [30], comprising 15 healthy volunteers (5 women; average age: 23.9 years, SD ± 2.3, age range: 19–29 years) to conduct decoding the simulations described in Section 2.5 “Simulation analysis”. In the current study (Study 2), we recruited twenty healthy participants (13 women; average age: 25.2 years, SD ± 3.1, age range: 20–30 years) through advertisements posted on the University of Geneva website. To be included in the study, participants had to exhibit no severe neurological or psychiatric conditions according to self-reports. The two groups included different individuals. Both studies were approved by the local Ethics Committee (Commission Cantonale d’Ethique de la Recherche, project 2022-00451) and were performed in accordance with the Declaration of Helsinki. All participants provided written informed consent and received financial compensation for their participation.

### 2.2. Experimental Paradigm

All participants of Study 2 underwent two BCI control sessions on different days, spaced at least 3 weeks apart to avoid potential learning effects, which could represent a confounding factor when evaluating differences in performance between the two sessions. During these sessions, either a static or an adaptive LDA classifier was used in counterbalanced order across participants. BCI control sessions were conducted in a room shielded optically, acoustically, and electrically, each of which lasted approximately 3 h, amounting to a total of 6 h of experimental time per participant. Participants were orally briefed about the overall procedure before the experiment began.

As in Study 1, the goal of Study 2 was to decode the imagery of two syllables, /fɔ/ and /gi/ (presented in written form on the screen during the experiment as “fo” and “ghi”). They were selected based on their distinct phonological and acoustical attributes concerning the consonant manner (fricative vs. plosive), place of articulation (labiodental vs. velar), vowel place (mid-back vs. high front), and rounding (rounded vs. unrounded). Previous research indicated that distinct phonetic features elicit different neural responses [11,50,51,52]. We assumed this choice would maximize the discriminability of the neural signals elicited by the imagery of the two syllables, hence facilitating decoding. 

Participants were instructed to engage in the imagery of one of the two syllables, concentrating on the kinesthetic sensations associated with actual pronunciation. They were instructed to avoid auditory or visual representations, such as internal hearing or visualizing the syllable in written form. Clear instructions were provided to refrain from any physical movements during the imagery process, with particular emphasis on avoiding facial movements and any subvocalization. They were instructed to maintain focus during the syllable imagery and were allowed to take breaks if needed.

### 2.3. EEG Acquisition and BCI Loop

#### 2.3.1. EEG Recording

EEG data were acquired using a 64-channel ANT Neuro system (*eego mylab*, ANT Neuro, Hengelo, The Netherlands) with a 512 Hz sampling rate. Electrode AFz served as the ground, and CPz was used as the reference. Impedances for the ground and reference electrodes were consistently maintained below 20 kΩ, while the impedances for all other electrodes were kept under 40 kΩ throughout the experiment. 

Neural signals were collected through the Lab Streaming Layer system (LSL, https://github.com/sccn/labstreaminglayer, accessed on 20 February 2024).

While the EEG was being recorded and the BCI was in operation, participants remained seated comfortably in front of a computer screen, with their hands resting on their lap.

#### 2.3.2. BCI Loop

The BCI loop was developed upon the *NeuroDecode* framework (Fondation Campus Biotech Geneva, https://github.com/fcbg-hnp/NeuroDecode.git, accessed on 20 February 2024), employed in previous BCI experiments [30,53,54]. 

The BCI experiment included three phases: (a) *offline training*, (b) *classifier calibration*, and (c) *online control*.

During the *offline training*, participants performed the syllable imagery without receiving real-time feedback while the data were being recorded. The offline session comprised 4 blocks of 20 trials in Study 1, and 3 blocks of 30 trials in Study 2, and lasted for 25–30 min. Within each block, both syllables were represented an equal number of times in randomized order. The experimental paradigm is shown in Figure 1. In both studies, each offline trial began with a text indicating the trial number (1 s), followed by a fixation cross (2 s), after which a written prompt indicated which of the two syllables the participants had to imagine pronouncing (2 s). After the cue disappeared, an empty battery appeared on the screen and immediately started to fill up until fully filled across the subsequent 5 s. Participants were instructed to start imagining pronouncing the cued syllable as soon as the battery appeared and to stop once the battery was full. The same visual feedback was used for both classes to minimize potential confounds that could arise from different sensory-afferent information associated with each of the two syllables (e.g., a different movement direction of the feedback). The choice of the battery ensured the feedback to be provided in a simple and intuitive way, to minimize the cognitive load associated with the visual stimulation and to interfere the least with the performance of the imagery. Participants were instructed to keep a consistent pace while imagining repeating the syllable throughout the entire experiment. Subsequently, participants were given a rest period while the word “Rest” (4 s) was displayed on the screen. 

During the *classifier calibration* phase, offline data were used to train an LDA classifier. A standard LDA classifier uses a hyperplane to separate data from different classes by finding the optimal projection that maximizes between-class distance while minimizing within-class distance [55]. The choice of this classifier was motivated by several studies showing the effectiveness of the adaptive version in improving BCI performance as compared to the static version [32,36,46,47,56,57]. In addition, an LDA classifier offers advantages such as a smaller dataset required for training the model and lower computational resources, resulting in faster computation compared to alternative methods such as deep learning. These characteristics were particularly relevant given the real-time updating of the adaptive classifier and one of the primary goals of our study being to reduce the experimental training time. This computationally parsimonious choice is also well suited to our relatively straightforward binary classification task.

The LDA classifier was designed to distinguish between the two syllables based on features derived from the EEG power spectral density (PSD) in the 1–70 Hz frequency range, with a 2 Hz interval, and from all EEG electrodes except the reference channel and the two mastoid electrodes (leaving 61 channels). Prior to feature extraction, a 50 Hz notch filter and Common Average Reference (CAR) were applied to the data. The PSD was computed using a 500 ms sliding window with a 20 ms overlap. The classifier’s performance was determined through an 8-fold cross-validation accuracy (“CV accuracy”).

During the *online control* phase, participants imagined pronouncing either one or the other syllable as in the *offline training* phase, except that this time their brain activity was analyzed in real time by the LDA classifier, and the output of the decoder was used to control the battery filling in real time. Participants were informed that the visual feedback would be updated (around every 50 ms) according to the classifier’s probability output: if the output corresponded to the cued syllable, the battery filling increased; otherwise, the battery filling decreased. The battery-filling real-time control continued until the battery was either fully charged or a 5 s timeout elapsed. Participants were instructed to use the same syllable-imagery strategy as during the *offline training*.

In Study 1, online visual feedback was provided using a static classifier; thus, the parameters obtained in the *classifier calibration phase* remained the same throughout the entire *online control* phase (see Figure 2a). In Study 2, in addition to an online control session employing the static LDA classifier, participants performed another session using the adaptive LDA classifier. During this session, the LDA classifier was updated iteratively using features derived from newly acquired data (Figure 2b) every time a new trial from both classes (i.e., the two syllables) became available.

### 2.4. Adaptive LDA Classifier

For a binary classification, the LDA operates on the principle of distinguishing between two classes by assuming that both follow a normal distribution. This assumption is applied to the offline dataset by retrieving the mean vectors and covariance matrices for each of the two classes. Furthermore, it is assumed that both classes are characterized by a common covariance matrix [55]. The LDA classifier can ascertain the data label using Equations (1)–(3). The value *D*(*x*) represents the distance of the feature vector *x* from the separating hyperplane, characterized by its normal vector *w* and bias *b*. The parameters *w* and *b* can be expressed in terms of the inverse covariance matrix Σ^−1^ (2) and the mean of each class *μ*_1_ and *μ*_2_ (3). In our case, the feature vector *x* is derived from one sliding window of neural data relative to the imagery of one syllable. The sign of *D*(*x*) determines the class (i.e., the syllable imagined) to which the observed feature vector *x* is assigned.
(1)$$D(x)=w^{T}\cdot x+b$$
(2)$$w=\Sigma^{-1}\cdot \left(\mu_{2}-\mu_{1}\right)$$
(3)$$b=-w^{T}\cdot \frac{1}{2}\cdot \left(\mu_{1}+\mu_{2}\right)$$

One way to achieve adaptive classification consists of updating the parameters *w* and *b* based on newly acquired data in real time, during the *online control* phase. To do so, we employ an algorithm that consists of changing at each time *t* (in our case, when each class has a new trial) the value of the mean *μ* of each class *i* (equal to 1 or 2, for a two-class problem) and the common inverse covariance matrix Σ^−1^, considering N data samples, according to Equations (4) and (5), respectively [47]. In these equations, an updated coefficient (*UC*) is used to set the respective contribution of the previous model’s values of the mean *μ_i_*(*t* − 1) and covariance matrix Σ(*t* − 1), as well as the current features *x*(*t*), and ultimately to update the current mean *μ_i_*(*t*) and covariance matrix Σ(*t*). The values of the *UC* (one for the mean *UC_μ_* and the other for the covariance matrix *UC*_Σ_), between 0 and 1, set the updating speed of the adaptive classifier. A value nearing zero indicates minimal influence of the new samples on the model, whereas a value approaching one results in a stronger reliance on newly acquired samples as compared to old ones and leads to a faster update.
(4)$$\mu_{i}\left(t\right)=\left(1-{UC}_{\mu }\right)\cdot \mu_{i}\left(t-1\right)+{UC}_{\mu }\cdot \;x_{i}\left(t\right)$$
(5)$$\Sigma \left(t\right)=\left(1-{UC}_{\Sigma }\right)\cdot \Sigma \left(t-1\right)+\frac{1}{N-1}{\cdot UC}_{\Sigma }\cdot (x_{i}\left(t\right)-\mu_{i}\left(t-1\right))\cdot {\left(x_{i}\left(t\right)-\mu_{i}\left(t-1\right)\right)}^{T}$$

The inverse of the covariance matrix Σ(*t*)^−1^ can be computed by applying the Woodbury matrix identity [58], stating that, for a given matrix *B*,
(6)$$B=A+UV^{T}$$
where *A*, *U,* and *V* are conformable variables, *^T^* indicates the transpose operator, and its inverse $B^{-1}$ can be obtained as follows: (7)$$B^{-1}={\left(A+UV^{T}\right)}^{-1}=A^{-1}-A^{-1}\cdot U\cdot {\left(I+V^{T}\cdot A^{-1}\cdot U\right)}^{-1}\cdot V^{T}\cdot A^{-1}$$

Combining Equations (5)–(7), we can represent Σ(*t*) as the matrix *B* (Equation (8)) and identify the other terms of Equation (7) as follows: (8)$$B=\Sigma \left(t\right)$$
(9)$$A=\left(1-{UC}_{\mu }\right)\cdot \Sigma \left(t-1\right)$$
(10)$$U=\frac{1}{N-1}{\cdot UC}_{\Sigma }\cdot (x_{i}\left(t\right)-\mu_{i}\left(t-1\right))$$
(11)$$V^{T}={\left(x_{i}\left(t\right)-\mu_{i}\left(t-1\right)\right)}^{T}$$

Using these equations and by integrating values from *x*(*t*), Σ(*t* − 1)^−1^, *μ_x_*(*t* − 1), *UC*_Σ_, and *UC_μ_*, the inverse of the covariance matrix Σ(*t*)^−1^ and *μ_i_*(*t*) for each class was updated iteratively in real time.

### 2.5. Simulation Analyses to Optimize the Adaptive LDA Classifier

While most parameters in the adaptive LDA are data-driven (e.g., *x*, Σ^−1^, *μ_x_*), *UC*_Σ_ and *UC_μ_* are constants that need to be pre-specified. To identify the optimal UC values, specific to the syllable imagery task, to be used in the adaptive LDA classifier, we conducted classification simulations on the EEG dataset acquired for Study 1 [30]. Importantly, the binary /fɔ/ vs. /gi/ syllable imagery task was exactly the same in Study 1 (optimization) and in Study 2 (static vs. adaptive comparison). The Study 1 dataset consisted of EEG recordings from 15 healthy participants who trained to control a speech BCI daily over five consecutive days, yielding a total of 75 datasets. 

We probed a series of values for the *UC_μ_* and the *UC*_Σ_, ranging from 0.4 × 2^−14^ to 0.4 in increments of powers of two (15 values in total). The upper bound was set to prevent the update speed from being excessively rapid. To identify the best *UC* value range, we evaluated the predicted accuracy (PA) obtained through simulations of classification using every possible combination of *UC_μ_* and *UC*_Σ_ values (UC pair), for each of these pre-recorded datasets. To compute the PA, an initial LDA classifier was constructed using data acquired during the *offline training* phase (80 trials). Then, for a given UC pair, the classifier was updated with data from the *online control* for each new trial and class. To classify each trial, we split the trial into 500 ms sliding windows, with a 50 ms overlap, simulating the real-time process during *online control*. The decoding accuracy for the cued syllable was computed for each sliding window, and probabilities from all decoding accuracies were averaged. The trial was labeled as correctly or wrongly classified based on the average value being above or below a 50% chance level. Finally, the PA for that specific UC pair was computed as the percentage of correct trials over the total number of trials. This led to a 15 × 15 matrix, the “adaptive predicted accuracy” (adaptive PA) matrix, in which the x-axis represents the range of possible values for the *UC*_Σ_ and the y-axis the range of *UC_μ_* values, and each element of the matrix indicates the adaptive PA.

To assess the effectiveness of this approach, we compared the adaptive PA with the PA obtained with a static LDA classifier. To compute the static PA, we apply an approach analog to the one described above: first, we computed a classifier based on the *offline training* dataset and subsequently used it to predict each trial in the *online control*, this time without updating the classifier parameters. We then considered the percentage of trials correctly classified to obtain the static PA for each of the 75 EEG datasets. Of note, differently from the adaptive PA matrix, the static classifier did not involve different *UC* values; thus, there was only one static PA value per dataset. 

#### Choice of the Optimal UC Pair

The adaptive PA matrix and the static PA were used to derive three new matrices, from which we selected the UC pair: the “*Modulated PA*”, the “*Consistency Score*”, and the “*Modulated Bias*” matrices. To compare the PAs obtained with the static and adaptive classifier and identify the optimal UC pair, we derived, for each dataset, the “*Modulated PA*” matrix by subtracting the static PA from the corresponding a-PA matrix, separately for each of the 75 datasets. We then computed the average of all “*Modulated PA*” matrices. Finally, we identify the UC pair (*UC_μ_* and *UC*_Σ_ values) that led to the highest PA increase with the adaptive as compared to the static classifier. 

Next, we evaluated the consistency of the effect of the adaptive classifier on PA for each UC pair across the 75 datasets. To assess this, we transformed each ”*Modulated PA*” matrix into a binary matrix by assigning a score of +1 to positive *Modulated PA* values and 0 to negative or null values. We then summed the binary matrices across all datasets to obtain a “*Consistency Score*” matrix, in which higher values indicate the number of datasets showing improved performance with the adaptive classifier as compared to the static one.

Non-stationarity affecting EEG signals often leads to the classifier being biased, that is, the erroneous tendency of the classifier to make predictions consistently towards one particular class. By constantly updating its parameters based on incoming signals, adaptive classifiers have been proven effective in reducing the classifier’s bias [33,59]. 

We evaluated which UC pair would lead to the lower bias, as follows: for each dataset, we considered the bias of the static classifier (static Bias) and the bias of the adaptive classifier (adaptive Bias) for each UC pair as the absolute value of the difference in the PA between the two classes (Equations (12) and (13)). Then, for each of the 75 EEG datasets, the individual “*Modulated Bias*” matrix was computed as the difference between the adaptive Bias and the static Bias. Last, we calculated the average of all the “*Modulated Bias*” matrices, yielding one average “*Modulated Bias*” matrix across all 75 datasets.
(12)$$static\;Bias=|static\;{PA}_{class1}-static\;{PA}_{class2}|$$
(13)$$adaptive\;Bias=|adaptive\;{PA}_{class1}-adaptive\;{PA}_{class2}|$$

Based on these three matrices, we identified one UC pair to be integrated in the adaptive classifier as the one that maximized the “*Modulated PA”* and the “*Consistency Score”* while minimizing the “*Modulated Bias”*. To quantitatively evaluate the validity of this choice, we conducted a two-tailed paired *t*-test to compare the static PA and the adaptive PA obtained from the pre-recorded datasets by using the select UC pair. We thus considered for each participant the average PA over the five days of training. 

### 2.6. Real-Time BCI Experiment to Evaluate the Adaptive Classifier’s Effectiveness 

We then used Study 2 to address our main hypothesis that the improvement obtained with the adaptive LDA classifier by the running simulation analysis should translate into better BCI control in real time. In this experiment, we employed the same UC pair as previously identified at the group level based on the Study 1 dataset. We divided our pool of participants into two groups based on the session they performed first, leading to an “Adaptive First” group and a “Static First” group.

The BCI experimental protocol was the same as described in the section “BCI loop” for both the static and adaptive classifier sessions, except that during the adaptive session, the classifier in the *online control* phase was updated continuously as described in Section 2.4. Every time a participant completed a new trial for each of the two classes (one per syllable), the classifier’s parameters *w* and *b* were re-calibrated using Equation (1) through (7), employing the optimal UC pair selected from the simulation analysis. This updated classifier was iteratively applied to the forthcoming data.

#### 2.6.1. BCI Control Performance

To evaluate performance during the real-time *online control* phase, we computed *BCI control performance* as the percentage of time where the classifier’s output matched the cued syllables, for each trial independently. Unlike the PA described in the previous sections, where each trial is labeled as correctly or incorrectly classified (hit/miss), the *BCI control performance* is calculated based on sliding windows of 500 ms and therefore can be considered a more fine-grained method of assessing performance. In addition, as the visual feedback is provided from the classifier’s output, the performance also reflects the subjective perception of control from the individual operating the BCI. 

Finally, the overall BCI control performance for each *online control* phase was determined by averaging values from all trials.

Differences in BCI control level achieved with the adaptive and with the static classifier were tested with two-tailed paired *t*-tests. To confirm there was no learning between the two sessions, we compared performances during the first and second sessions, irrespective of the order of the classifier used. Last, we compared the performance obtained with the adaptive and the static classifier separately for the “Adaptive First” and “Static First” groups.

Last, we investigated whether the improvement obtained with the adaptive classifier vs. the static one (irrespective of which classifier was used first) could be related to initial BCI performance. More specifically, we hypothesized that those participants who benefited the most from the adaptive classifier were those who exhibited better BCI control performance during the first session. To test this, we considered the difference in BCI control performance between the adaptive and static classifiers and computed Pearson’s correlation coefficient with the BCI control performance observed in the first session.

#### 2.6.2. Cross-Validation Accuracy

To evaluate differences in decoding accuracies between the *offline training* and *online control*, we computed a static LDA classifier separately for the offline and online datasets from both Study 2 experimental recordings (adaptive and static classifier) and tested the accuracy with an 8-fold cross-validation (CV). We checked that there was no difference in the CV accuracy between the two sessions during the *offline training*, as the experimental conditions were the same. We then tested differences in CV accuracy between the *offline training* and *online control*, as a previous study using the same paradigm highlighted a higher decoding accuracy when visual feedback was provided [30]. Last, we investigated whether the magnitude of this difference depended on the classifier used (static vs. adaptive). Each of these analyses assessing differences in CV accuracies employed two-tailed paired *t*-tests.

#### 2.6.3. Post Hoc Simulation Analysis and Individual UC Pair Optimization

To assess the reliability of our approach in identifying the optimal UC values, we performed the same simulation analysis as used for the pre-recorded Study 1 dataset (Section 2.5) on the Study 2 dataset. In particular, we evaluated the overlap in the three distinct PA metrics (namely the “*Modulated PA*”, the “*Consistency Score*”, and the “*Modulated Bias*”) used to choose the optimal UC pair between the two datasets.

Using the online BCI dataset, we further tested the difference in PA obtained for the static vs. adaptive classifier using the selected UC pair as described in Section 2.5 (two-tailed paired *t*-tests).

Previous studies have pointed out that the speed of adaptation is a crucial element for the adaptive classifier, as a classifier updating too often might lead to instability in the visual feedback, whereas a classifier updating too slowly might fail to effectively track neural changes due to learning [60], both causing the co-adaptive system to underperform [61]. To further optimize the UC parameters, an alternative approach consists of tuning the UC pair for each individual separately, rather than applying the values computed at the group level. Such an approach would, however, require a considerable amount of data and would be computationally time-consuming. We therefore decided not to explore this option in real time, but we evaluated its validity for future developments through a simulation analysis. 

We used the same simulation analysis method as detailed in Section 2.5, but this time we based the selection of the UC pair exclusively on the “*Modulated PA*” matrix, considering, for each participant, data from both sessions. For each participant, we selected the UC pair that led to the strongest increase with the adaptive classifier (i.e., the highest value in the “*Modulated PA*”) and tested for differences with PAs obtained with the static and adaptive classifier with the UC pair obtained at the group level (one-way repeated measures ANOVA). Post hoc comparisons were carried on with a two-tailed paired *t*-test. 

## 3. Results

### 3.1. Simulation Analyses to Optimize the Adaptive LDA Classifier

First, we identified the optimal *UC*_Σ_ and *UC_μ_* values (i.e., a UC pair) to be used during real-time BCI control by performing simulation analysis on a previously recorded dataset employing the same experimental paradigm as the one tested in the present study [30]. We based the choice of the UC pair on three distinct metrics: “*Modulated PA*”, “*Consistency Score*”, and “*Modulated Bias*”. The “*Modulated PA*” matrix, obtained as the difference in PA between static and adaptive classification, indicated that combinations yielding the strongest increase in PA (amounting to 5%) with the adaptive classifier fall within a restricted *UC* range (*UC_μ_*: 0.4 × 2^−6^ to 0.4 × 2^−4^; *UC*_Σ_: 0.4 × 2^−4^ to 0.4 × 2^−1^; Figure 3a).

Next, we considered the consistency of the adaptive classifier to improve PA and found that the above-mentioned UC range based on the “*Modulated PA*” had the highest consistency score, with 54 out 75 datasets showing an improvement (*UC_μ_*: 0.4 × 2^−6^; *UC*_Σ_: 0.4 × 2^−3^ to 0.4 × 2^−1^; Figure 3b). Based on both the “*Modulated PA*” and “*Consistency Score*”, we selected as the optimal *UC* parameters for the adaptive LDA classifier to be tested online, a *UC_μ_* equal to 0.4 × 2^−6^ and a *UC*_Σ_ at 0.4 × 2^−3^ (red dots in Figure 3a,b) and then verified that the chosen parameters were also effective in decreasing the bias between the two classes. Accordingly, the optimized values used with the adaptive classifier allowed a bias reduction of 12.5% as compared to the static classifier (“*Modulated Bias*”, Figure 3c).

We then quantified whether the decoding improvement due to the adaptive LDA classifier was statistically significant across participants and found that this was the case (T_14_ = 4.29, *p* < 0.01, d = 1.1, Figure 3d). Out of 15 participants, 13 exhibited a higher PA with the adaptive LDA classifier than with the static classifier.

### 3.2. Real-Time BCI Experiment

#### 3.2.1. BCI Control Performance

We tested the effectiveness of using the selected UC pair in a real-time setting with a new group of participants, who performed two separate BCI control sessions, one employing a static LDA and the other an adaptive LDA classifier, in randomized order. Two-tailed paired *t*-tests revealed higher BCI control performance with the adaptive than the static classifier (T_19_ = 2.99, *p* < 0.01, d = 0.67, Figure 4a). This effect was present in 15 out of 20 participants and did not depend on the order in which each of the classifiers was used (“Adaptive First” and “Static First” groups). Both groups had a statistically significant improvement with the adaptive classifier (“Adaptive First” group: 54.3 ± 1.56% for the adaptive classifier, 52.3 ± 2.11% for the static classifier, T_9_ = 2.27, *p* < 0.05, d = 0.72; “Static First” group: 57.2 ± 2.31% with the adaptive classifier, 52.3 ± 0.35% for the static classifier, T_9_ = 2.34, *p* < 0.05, d= 0.74). 

These results confirmed our initial hypothesis that the adaptive classifier allowed for a better BCI control as compared to the static one. 

Furthermore, we also controlled for learning across sessions and found no difference in BCI control between the first (53.3 ± 0.81%) and second session (54.7 ± 1.62%, T19 = 1.07, *p* = 0.3, d = 0.24, Figure 4b).

To test whether participants with better initial performance benefited more from the adaptive classifier than the poorer performers, we correlated the difference in BCI performance between the adaptive and static classifier with the BCI performance of the first session and found no statistically significant relationship (Figure 4c, ρ = −0.138, *p* = 0.56). However, this statistical result was driven by one outlier point, which had the highest performance of all recordings with the static LDA classifier (static: 70.9%; adaptive: 67.8%). Upon removing this value, the correlation coefficient reached statistical significance (Figure 4d, ρ = 0.458, *p* = 0.048).

#### 3.2.2. Cross-Validation Accuracy

As expected, we found no difference in the cross-validation (CV) accuracy obtained by classifying data from the *offline training* phase between the adaptive (49.7 ± 0.72%) and the static classifier (50.4 ± 0.88%) (T_19_ = 0.59, *p* = 0.56, d = 0.13, Figure 5a). Surprisingly, there was no significant difference in the CV accuracy for the *online control* phase (Figure 5b, T_19_ = 1.4, *p* = 0.18, d = 0.31). We then tested for a decoding improvement from the *offline training* to the *online control*, separately for the adaptive and static sessions, and found an increase for both sessions (Static classifier: offline CV = 50.4 ± 0.88%, online CV = 53.9 ± 1.7%, T_19_ = 2.95, *p* < 0.01, d = 0.66; Adaptive classifier: offline CV = 49.7 ± 0.72%, online CV = 56 ± 1.95%, T_19_ = 3.11, *p* < 0.01, d = 0.7; Figure 5c,d). The critical test was whether the improvement in the online setting was potentiated by the adaptive classifier, i.e., whether there was an added value of an adaptive classifier in a closed loop setting. We found a trend for a stronger increase associated with the use of the adaptive classifier (Adaptive: 6.3 ± 2%, Static: 3.5 ± 1.18%, T_19_ = 1.85, *p* = 0.08, d = 0.41, Figure 5e). 

#### 3.2.3. Post Hoc Simulation Analysis

We then investigated whether the selected UC pair used for the adaptive LDA classifier and derived from the pre-recorded dataset aligned with that obtained from the real-time experiment. We applied the same simulation analyses as described in Section 2.5. to compute the “*Modulated PA*”, “*Consistency Score*”, and “*Modulated Bias*” for this new dataset. The selected UC pair fell within optimal PA regions of the real-time experiment. More specifically, the selected UC pair was the second pair in terms of increased accuracy with the adaptive classifier as compared to the static classifier (“*Modulated PA*”, Figure 6a). Such an improvement using the selected PA appeared in 26 out of 40 datasets, as indicated in the “*Consistency Score*” matrix, close to the maximum value of 29 datasets (Figure 6b, black square). Of note, the UC pair that led to the highest consistency value was associated with a lower improvement in PA than the UC pair used in the real-time experiment (red square in the “*Modulated PA*” matrix). The selected UC pair also reduced the bias between the two classes, by −10.92%, the second strongest reduction throughout the entire “*Modulated Bias*” matrix (Figure 6c). 

Two-tailed paired *t*-tests showed an improvement in PA using the adaptive classifier with the selected UC pair relative to the static classifier, consistent with the results from the same simulation analysis conducted on the previously acquired Study 1 dataset (Figure 6d, T_(19)_ = 3.2, *p* < 0.01, d = 0.72). This overlap in the updating parameters obtained from the two studies shows that the method we propose is not only effective but also reliable.

### 3.3. Individual UC Pair Optimization

Finally, we investigated whether adapting the UC pair for each participant further boosted PA. A one-way repeated measures ANOVA with the classifier type as a factor (static or adaptive, with the UC pair optimized at the group level, and adaptive with the UC pair optimized at the individual level) showed a significant main effect (F_2,38_ = 43.8, *p* < 0.001, η^2^_p_ = 0.166). Post hoc paired *t*-tests with Bonferroni corrections revealed that the PA obtained with the individual UC pair was statistically significantly higher than the static classifier (T_19_ = 8.72, Bonferroni-adjusted *p* < 0.001, d = 1.95) and the group UC pair (T_19_ = 8.65, Bonferroni-adjusted *p* < 0.001, d = 1.93, Figure 6d).

## 4. Discussion

In this study, we explored whether employing an adaptive LDA classifier for decoding the imagery of two syllables from EEG signals would yield improved BCI control compared to a static classifier, commonly used in the field. To do this, we calibrated the adaptive LDA based on simulation analyses performed on a pre-recorded dataset and then tested its effectiveness in a new experiment in which healthy volunteers performed two sessions of BCI control, one employing a static LDA and the other an adaptive LDA classifier, over two separate days. 

As hypothesized, we found that the adaptive classifier enhanced BCI control performance in the real-time BCI experiment (Figure 4a). This effect was not related to participants learning to operate the BCI system, as no difference was found between the first and the second experimental session, and it was present irrespective of the classifier used during the first session. These findings support the benefit of using adaptive classifiers in the field of imagined speech decoding, which is notoriously characterized by low online decoding accuracy, barely exceeding the chance level [18,30,62]. 

Furthermore, we found no difference in decoding accuracy computed in post-processing (i.e., CV accuracy) between the static and the adaptive session, neither for the *offline training* nor for *online control* datasets. This result was expected for the *offline training* (Figure 5a), the first part of the experiment preceding the classifier training when no feedback is provided to the user (thus characterized by the same experimental conditions). However, it was surprising to observe no difference for the *online training* dataset (Figure 5b): since we measured higher BCI control performance when computed in real time using the adaptive classifier, it would have been plausible to find such a difference also when decoding was applied in post-processing using a different approach. This discrepancy suggests that the improvement observed in real time is solely due to an optimization on the decoder side, and it is not accompanied by an adaptation of the underlying neural activity when using one or the other classifier. If a difference had emerged along the course of the experiment, it would have likely been due to the neural data being more discriminable (between the two classes) during the adaptive session. Ultimately, this would be reflected in a higher CV accuracy. 

Confirming previous results using the same paradigm [30], we found higher decoding accuracy for signals recorded during the *online control* than the *offline training* (Figure 5c,d), indicating that the real-time feedback helped users to self-regulate brain patterns, in line with Study 1 [30]. Interestingly, the extent of the decoding improvement when the imagery task was performed within a closed loop (*online control*) tended to be stronger in the adaptive session than in the static one (Figure 5e). This suggests that, if not over the course of a single *online control* session, boosting real-time control with an adaptive classifier has the potential to enhance neural discriminability in the medium term. We speculate that these neural differences would affect the most discriminant features for the syllable decoding, similar to the effect of the adaptive classifier in enhancing sensorimotor rhythms in motor-imagery-based BCIs [37,63,64].

Enhanced benefits from the real-time feedback with an adaptive classifier might also accelerate learning to operate the BCI. The acquisition of BCI control skills has been shown to be possible using a static classifier by training participants over 5 consecutive days, using the same experimental paradigm [30]. Similarly, using an adaptive classifier, a previous study employing a BCI based on motor imagery demonstrated an improvement in performance over 3 days [57]. We can thus reasonably expect that an adaptive classifier further promotes learning by capitalizing on the co-adaptation between the decoders and the user, previously shown to induce neural changes on a short time scale [65]. It remains to be addressed to what extent learning dynamics and the associated neural changes elicited by training across multiple sessions differ when using a static or an adaptive classifier. 

Improving the accuracy of current decoders could further boost metacognitive abilities that play a key role in learning to consciously self-regulate ongoing brain activity [66,67]. This conscious aspect, together with an unconscious type of learning, originally posited by Lacroix as the dual-process model [68], has been proposed as a theoretical framework to account for learning BCI control [67]. Similarly, improving decoding will impact higher-order mechanisms related to action execution, such as the subjective sense of being in control (i.e., the sense of agency) of BCI actions. This fundamental mechanism has been characterized for motor imagery BCI control as relying mainly on the congruency between intentions and sensory feedback [69,70]. By enhancing these higher-order cognitive mechanisms, a more accurate decoding from the machine side could further improve BCI control on the user side in a synergistic fashion. 

Our experimental results also show that the approach we used to identify the optimal update parameters at the group level is robust and reliable, as we found matching values using the pre-recorded dataset and the real-time dataset acquired specifically to validate our method. Albeit effective, we found that the *UC* values chosen at the group level did not benefit all participants, as 5 out of 20 did not show improved BCI control with the adaptive classifier. We hypothesized that this difference might be due to the chosen UC values not being suitable for certain participants. Previous studies have suggested that sub-optimal updating parameters could hinder the co-adaptation between the user and the system [33,34,60,61]. We then tested whether optimizing the *UC* values at the individual level resulted in a more uniform increase and found that individualizing the *UC* parameters improved PA in all participants. A possible speculation is that participants who do not benefit from the UC chosen at the group level might present less neural activation during the imagery due to a lower ability to self-regulate brain patterns. In this direction, the lack of skills to effectively modulate the sensorimotor rhythms has been associated with reduced improvements, despite the use of an adaptive classifier in previous motor-imagery-based BCI studies [56,63]. Further work will investigate whether such neural differences associated with the ability to exploit the full potential of an adaptive classifier exist also for speech imagery. 

Furthermore, the fact that the adaptive classifier tested in real time was not optimally tailored to each participant’s individual characteristics might explain the absence of a clear link between initial BCI control abilities and the improvement associated with the use of an adaptive classifier that we initially hypothesized. In fact, we found such a positive trend to be present but heavily affected by one outlier who performed surprisingly well with the static classifier. Although promising, it should be noted that simulations to individualize UC values require substantial amounts of data and computational resources; thus, their effectiveness remains to be established in real-time control settings. 

Another potential optimization of the adaptive classifier consists of varying the *UC* values over the course of the experiment instead of being kept constant as in our study. Previous studies have put forward the use of Kalman adaptive LDA classifiers, whereby the Kalman gain would vary the *UC* values according to the classifier’s output in real time in a supervised fashion [32,33]. More specifically, if the classifier’s output is accurate, the system will rely more on the current decoding model, slowing the update speed. Conversely, if the classifier’s predictions are inaccurate, the update speed will be increased [46,47,71]. This approach could constitute a further improvement of our decoder, together with integrating a feature selection step that would focus the decoding on specific regions (left central and temporal bilateral) and frequency bands (alpha to low-gamma), shown to be the most discriminant with surface EEG for syllable imagery [30]. Future directions will also explore the use of other classifiers suitable for adaptive decoding that might offer performance superior to the LDA classifier. The Adaptive Support Vector Machine (SVM) is one candidate, shown to outperform LDA for motor imagery decoding [72]. In the context of imagined speech, static SVMs have been previously employed for offline simulations [27,73] and in one of the few online studies [18]. Last, it is worth noting that semi-supervised or unsupervised classifiers might be more effective in shortening training time [74], increasing the ecological validity of our BCIs, and boosting co-adaptation [36,56].

## 5. Conclusions

The present study shows for the first time the effectiveness of using an adaptive classification approach to improve BCI control based on imagined speech from surface EEG, notoriously known to be affected by low performance. Such technical efforts might appear marginal as compared to recent advances in decoding attempted speech from intracranial implants, with rates approaching half of normal speech [4,6]. While non-invasive systems will presumably never constitute a means of communication, simple yet effective binary BCIs can in fact be instrumental in enabling users to learn self-regulating brain patterns in preparation for operating more complex BCI systems. This approach has been shown in the context of an intermediate phase that forms a part of progressive training, viewed from the standpoint of human learning [75]. Using such a progressive training method, previous studies achieved high BCI performance in motor imagery tasks [76,77,78,79], and the learning previously observed with this paradigm [30] suggests this approach could be applied to BCI based on imagined speech.

Moreover, the use of adaptive classifiers is particularly relevant for surface-EEG recordings as an effective means to increase the number of decoding classes. Most of the current BCIs are limited to binary classification due to the necessity of having enough training data to counterbalance the low signal-to-noise ratio. By using data from the real-time control as training data, an adaptive classifier shortens the training time and favors the development of multi-class BCIs [74,80]. Given their superior decoding performance on the machine side and the potential implication of fostering engagement and learning to self-regulate brain patterns on the user side, adaptive classifiers offer a time-saving approach to improve BCI controllability. 

In summary, our results encourage the systematic use of adaptive decoders in the field of speech imagery BCI, a compelling, robust, and easy-to-use technological solution that presents numerous advantages without any additional computational costs. 

## Figures and Tables

**Figure 1 brainsci-14-00196-f001:**
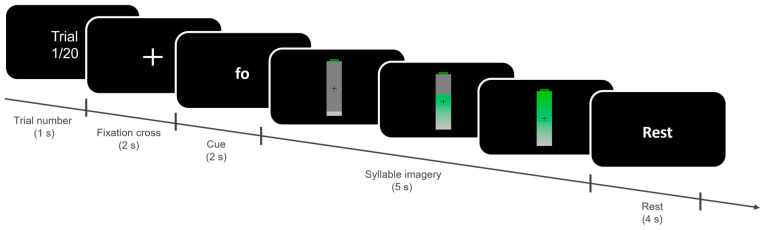
Experimental paradigm (Studies 1 and 2). Each trial began with the display of the trial number (1 s), followed by a fixation cross (2 s). Participants then were presented with a written cue (2 s) indicating the syllable they had to imagine (“fo” or “ghi”). Subsequently, an empty battery appeared on the screen, and participants were to start imagining pronouncing the syllable for 5 s, during the *offline training* phase or until the battery was filled in the *online control* phase. Participants were given 4 s to rest at the end of each trial while the text “Rest” appeared on the screen.

**Figure 2 brainsci-14-00196-f002:**
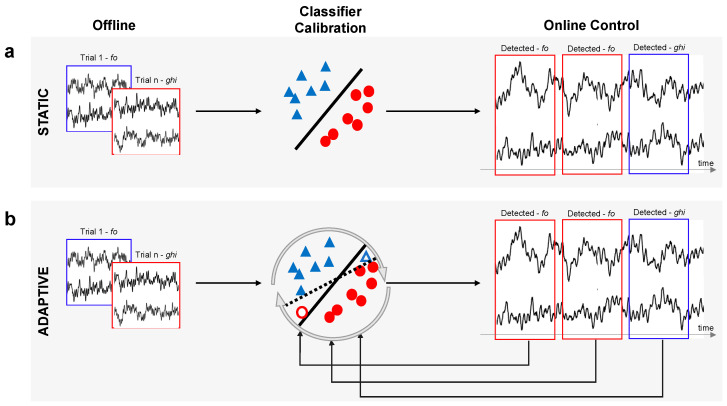
Schematic representation of the static and adaptive classifier. (**a**) Static: the LDA classifier was built based on the dataset acquired during the *offline training* phase and remained unchanged throughout the entire *online control* phase. (**b**) Adaptive: an initial classifier was built as in (**a**) and subsequently updated iteratively during the *online control* phase using features derived from new incoming data in real time. Red circles and blue triangles indicate features belonging to the two classes; empty shapes denote features from newly acquired data (during the *online control* phase) that are used to update the classifier. The solid lines in the *classifier calibration* step represent the separating hyperplane obtained with the initial static classifier, while the dashed line illustrates the updated classifier after the new data are integrated.

**Figure 3 brainsci-14-00196-f003:**
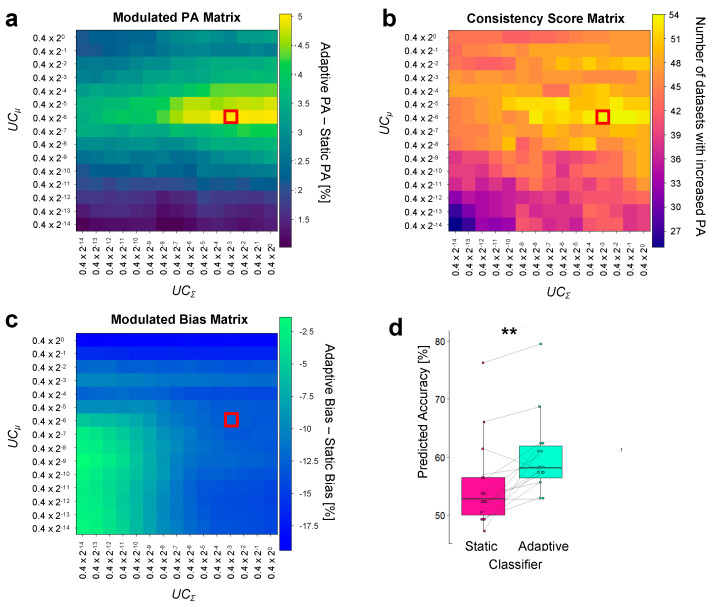
Simulation analyses to optimize the parameters for the adaptive LDA classifier. (**a**–**c**): Results from three metrics considered to evaluate the predicted accuracy using different UC pair values (*UC*_Σ_ and *UC_μ_*) using a pre-recorded dataset (Study 1). The x and y axes respectively depict the candidate value range for *UC*_Σ_ and *UC_μ_*, spanning from 0.4 × 2^−14^ to 0.4, comprising 15 distinct values. The red square indicates the selected UC pair. (**a**) “*Modulated PA*” matrix: values (percentage) in this matrix represent the difference between the adaptive PA and the static PA for each UC pair combination, averaged across datasets. The red square indicates the maximum value of PA improvement equal to 5% and obtained with *UC_μ_* = 0.4 × 2^−6^, and *UC*_Σ_ = 0.4 × 2^−3^. (**b**) “Consistency Score” matrix: each element of the matrix indicates the number of datasets where the adaptive PA is higher than the static PA for a given UC pair. The red square in correspondence with the UC pair chosen based on the “*Modulated PA*” highlights a value of 54, denoting that these specific UC parameters yielded improvements in 54 of the 75 datasets. (**c**) “*Modulated Bias*” matrix: each element of the matrix (percentage) is obtained as the difference in PA between the two classes (i.e., the two syllables) associated with each UC parameter combination. The highlighted value of −12.5% indicates that the PA disparity between the two classes was reduced by 12.5% compared to the static classifier. (**d**) Comparison of the average PA across the 5 datasets (in %) between static and adaptive classifiers with the selected UC pair (*UC_μ_* = 0.4 × 2^−6^, and *UC*_Σ_ = 0.4 × 2^−3^, indicated by the red square). The PA obtained was significantly higher for the adaptive (cyan, 60.1 ± 1.74%) than for the static classifier (pink, 55.1 ± 1.98%). Boxes represent the interquartile range (IQR), with the horizontal line indicating the median, and whiskers extending to data points that are within 1.5× the IQR from the upper and lower quartile. Individual points represent data from a single participant, and gray lines connect data points from the same participant. Significance is denoted with ** for *p* < 0.01.

**Figure 4 brainsci-14-00196-f004:**
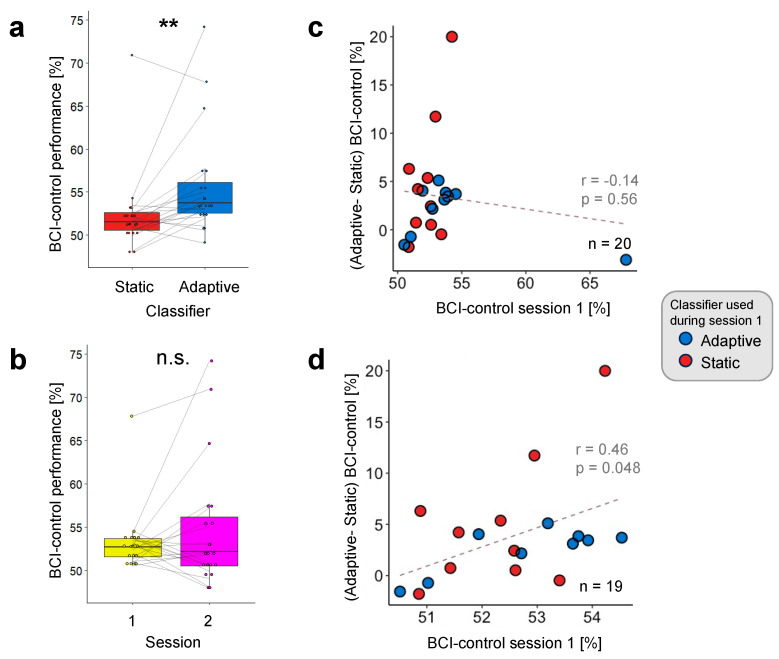
Impact of the adaptive LDA classifier on BCI control performance. (**a**) The adaptive LDA classifier led to better BCI control performance (55.7 ± 1.39%) relative to the static classifier (52.3 ± 1.04%). This effect was present in 15 out of 20 participants. (**b**) There was no significant difference in BCI control performance between the first (53.3 ± 0.81%) and the second session (54.7 ± 1.62%), irrespective of the classifier used. Individual dots represent individual BCI control performance; gray lines link data from the same participant. (**c**,**d**) Correlation between BCI control performance as observed during the first session (*x*-axis) and difference in performance between adaptive and static classifiers (*y*-axis). Dots indicate data from individual participants who used either the adaptive (blue) or static (red) classifier in the first session. The gray dotted lines illustrate the regression line. Results are shown considering data from all participants (**c**) and with one outlier removed (**d**). Significance is denoted with ** for *p* < 0.01.

**Figure 5 brainsci-14-00196-f005:**
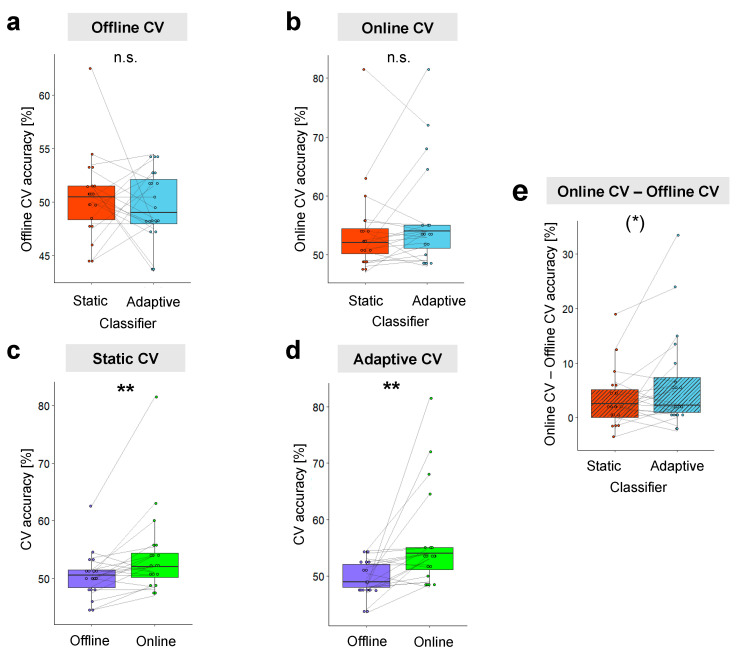
Cross-validation accuracy comparisons. We considered the cross-validation accuracy of decoding models obtained from a static classifier to compare decoding from different datasets (offline vs. online, static vs. adaptive). (**a**) There was no significant difference in the CV accuracy obtained from the *offline training* phase between the static (orange, 50.4 ± 0.88%) and the adaptive classifier (light blue, 49.7 ± 0.72%) sessions, (**b**) nor for the *online control* phase (static: orange, 53.9 ± 1.7%; adaptive classifier, light blue: 56 ± 1.95%). (**c**,**d**) Data acquired during the *online control* phase (i.e., when the user received real-time feedback, green) led to significantly higher CV accuracy than during the *offline training* phase for both the static classifier session (**c**) (online: 53.9 ± 1.7%; offline CV: 50.4 ± 0.88%) and the adaptive classifier session (**d**) (online: 56 ± 1.95%; offline: 49.7 ± 0.72%). (**e**) We tested for differences between the static and adaptive BCI sessions in the amount of CV accuracy improvement from the *offline training* to the *online control* phase and found a trend toward significance (adaptive: 6.3 ± 2%; static: 3.5 ± 1.18%). Dots represent individual participants’ data. Gray lines link data from the same participant. Significance is denoted with ** for *p* < 0.01 and (*) for *p* = 0.08.

**Figure 6 brainsci-14-00196-f006:**
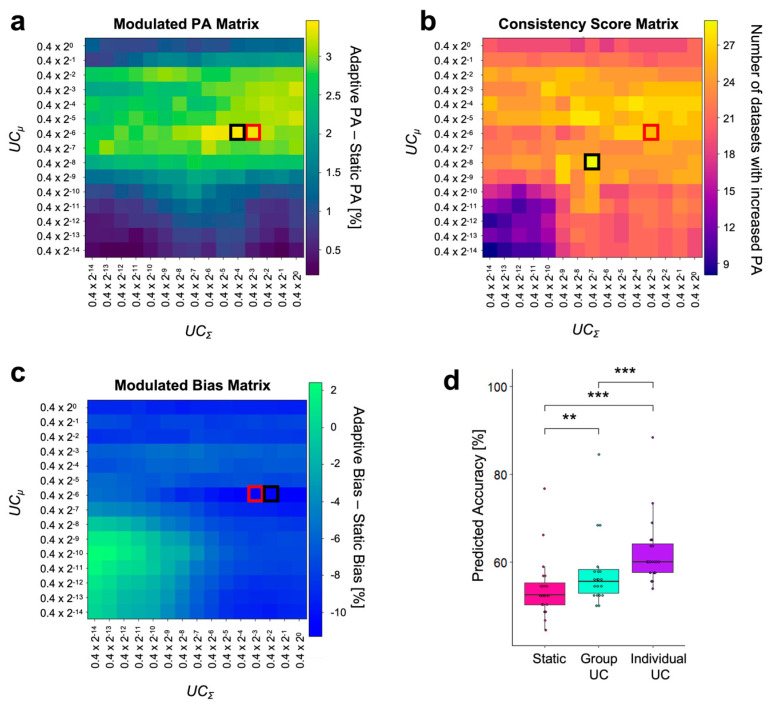
Simulation analyses of the real-time experimental dataset. (**a**–**c**) Predicted accuracy for different UC values computed with classification simulation on the real-time experimental dataset (Study 2). The x and y axes indicate the candidate range for *UC*_Σ_ and *UC_μ_*, respectively, spanning from 0.4 × 2^−14^ to 0.4, comprising 15 distinct values. Red squares represent the selected UC parameters used in the real-time experiment (selected UC pair: *UC_μ_* = 0.4 × 2^−6^, UC_Σ_ = 0.4 × 2^−3^). The black square represents the local maximum/minimum value in each matrix. (**a**) “*Modulated PA*” matrix: by using the selected UC pair (red square), the PA increased by 3.36% compared to the static classifier’s PA. The maximum increase was 3.45% (black square). (**b**) “*Consistency Score*” matrix: the selected UC pair (red square) led to a PA improvement in 26 out of the 40 datasets. The maximum value in this matrix was 29 (black square). (**c**) “*Modulated Bias*” matrix: the selected UC pair was effective in decreasing the bias between the two classes by 10.92%, and the maximum bias reduction was −11.14% (black square). (**d**) Comparison of simulated analysis performance across different classifiers. Each point marks the PA of an individual participant. A one-way repeated measures ANOVA confirmed a significant main effect of the classifier type on the PA. Subsequent post hoc paired *t*-tests showed that the PA for the adaptive classifier with individual *UC* (62.2 ± 1.73%, purple) was higher than both the static classifier (54.1 ± 1.59%, pink) and the adaptive classifier with Group *UC* (57.5 ± 1.8%, green). Performance with the adaptive classifier with group *UC* exceeded the PA obtained with the static classifier. Individual points represent data from a single participant. Significance is denoted with ** for *p* < 0.01 and *** for *p* < 0.001.

## Data Availability

The data that support the findings of this study are available from the corresponding author upon reasonable request. The data are not publicly available due to privacy reasons.

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
