# Peer review of "Adaptive LDA Classifier Enhances Real-Time Control of an EEG Brain–Computer Interface for Decoding Imagined Syllables"

_brainsci, 2024, doi:10.3390/brainsci14030196_

Round 1
Reviewer 1 Report
Comments and Suggestions for Authors
As a result of the review, the comments about the study are as follows;
• The title can more clearly reflect the main topic of the article. Try to be more specific.
• The abstract can present the main findings and important conclusions of the study more emphatically.
• The introduction can more effectively reveal the motivation and context of the study. State more clearly the research question you are interested in and how this study answers that question.
• The methods section may contain more detailed information on the data set used, measurements, experimental design, and statistical analysis methods.
• When describing the experiment protocol, you can include more information about participant selection, the process by which the experiment was conducted, and the data collection process.
• In the results section, the findings can be presented more concisely. Focus on the statistical analysis results and highlight important findings.
• In the discussion section, the relationship of the findings with the literature can be discussed in more depth. Comparisons can be made with previous studies.
• Limitations of the study and suggestions for future research could be stated more specifically.
• Visual elements (figures, tables) can be made more descriptive and understandable.
• Writing equations in a more orderly and aligned manner will increase readability.
• The way references are cited in the text should be corrected. Citation should be given in accordance with the journal template.
• Some sentences may be complex or too technical. Try to use simpler and more understandable language.
• Check for grammar and spelling errors.
Comments on the Quality of English Language
Moderate editing of English language required
Author Response
- The title can more clearly reflect the main topic of the article. Try to be more specific.
We would like to start by thanking the Reviewer for the comments. We propose to replace the word “speech” with “syllables” in the current version of the title. The revised version is the following: “Adaptive LDA classifier enhances real-time control of an EEG Brain-computer interface for imagined-syllable decoding”. The title contains all keywords of the current study, reflecting all main topics of the article. We believe that further extending the title length would contravene the concision requirements which is mentioned in Brain Sciences submission guidelines.
- The abstract can present the main findings and important conclusions of the study more emphatically.
We have emphasized the key finding in the abstract and extended the conclusions as follows (page 1, lines 26-32):
“As hypothesized, the adaptive classifier led to better performances than the static one in this real-time BCI-control task. Furthermore, the optimal parameters for the adaptive classifier were closely aligned in both datasets acquired using the same syllable imagery task. These findings highlight the effectiveness and reliability of adaptive LDA classifiers for real-time imagined speech decoding. Such an improvement can shorten the training time and favor the development of multi-class BCIs, representing a clear interest for non-invasive systems notably characterized by low decoding accuracies.”
3. The introduction can more effectively reveal the motivation and context of the study. State more clearly the research question you are interested in and how this study answers that question.
Following the Reviewer’s comment, we have highlighted the research question in the introduction as follows:
“Thus, we conducted a BCI-control study to test the hypothesis that an adaptive LDA classifier rather than a static LDA classifier would be effective in enhancing the control of a binary speech BCI based on the imagery of two syllables from EEG signals.”
We have also stated the motivation of the study at the end of the introduction (page 3, lines 122-123):
“This research addresses the limited effectiveness of current approaches to decode imagined speech in real-time, an important challenge that deserves prompt solutions.”
4. The methods section may contain more detailed information on the data set used, measurements, experimental design, and statistical analysis methods.
We have added more detailed information in the method section, also taking into consideration comments from other reviewers. As Reviewer 3 pointed out this section is long, we were particularly careful in extending it parsimoniously. We added information concerning the experimental paradigm as follows:
2.1 Participants
“In the current study (Study 2), we recruited twenty healthy participants (13 women; average age: 25.2 years, SD ±3.1, age range: 20-30 years) through advertisements posted on the University of Geneva website. To be included in the study, participants had to exhibit no severe neurological or psychiatric conditions according to self-reports.” (page 3, lines 118-121)
2.3.2 BCI loop
“The same visual feedback was used for both classes to minimize potential confounds that could arise from different sensory -afferent- information associated with each of the two syllables (e.g. a different movement direction of the feedback). The choice of the battery ensured the feedback to be provided in a simple and intuitive way, to minimize cognitive load associated with the visual stimulation and to interfere the least with the performance of the imagery.” (page 4, lines 179-184)
“In addition, a LDA classifier offers advantages such as a smaller dataset required for training the model and lower computational resources, resulting in faster computation compared to alternative methods such as deep learning. These characteristics were particularly relevant given the real-time updating of the adaptive classifier and one of the primary goals of our study being to reduce the experimental training time. This computationally parsimonious choice is also well-suited to our relatively straightforward binary classification task.” (page 5, lines 202-208)
“Prior to feature extraction, a 50 Hz notch filter and Common Average Reference (CAR) were applied to the data.” (page 5, lines 212-213)
“In Study 1, the online visual feedback was provided using a static classifier, thus the parameters obtained in the classifier calibration phase remained the same throughout the entire online control phase (see Figure 2a). In Study 2, in addition to an online control session employing the static LDA classifier, participants performed another session using the adaptive LDA classifier. During this session, [...]” (page 5, lines 226-230)
Figure 1: Experimental paradigm (Studies 1 & 2). Each trial began with the display of the trial number (1s), followed by a fixation cross (2s). Participants then were presented with a written cue (2s) indicating the syllable they had to imagine (“fo” or “ghi”). Subsequently, an empty battery appeared on the screen and participants had to start imagining pronouncing the syllable for 5 seconds, during the offline training phase, or until the battery was filled in the online control phase. Participants were given 4 seconds to rest at the end of each trial while the text “Rest” appeared on the screen. (page 6, lines 234-242)
We also provide additional information concerning equations (6) and (7). Concerning the statistical analysis methods, we used classical parametric statistics (repeated measurement ANOVA and two-tailed paired t-test) that are already reported completely in the original version of the manuscript.
- When describing the experiment protocol, you can include more information about participant selection, the process by which the experiment was conducted, and the data collection process.
As reported in the previous comment, we have provided more detailed information concerning the experimental approach, data collection, and the participants' selection:
“[...] through advertisements posted on the University of Geneva website. To be included in the study, participants had to exhibit no severe neurological or psychiatric conditions according to self-reports.” (page 3, lines 118-121)
“In Study 1, the online visual feedback was provided using a static classifier, thus the parameters obtained in the classifier calibration phase remained the same throughout the entire online control phase (see Figure 2a). In Study 2, in addition to an online control session employing the static LDA classifier, participants performed another session using the adaptive LDA classifier. During this session, [...]” (page 5, lines 226-230)
- In the results section, the findings can be presented more concisely. Focus on the statistical analysis results and highlight important findings.
We have emphasized the primary findings of our study in the results section as follows:
“These results confirmed our initial hypothesis that the adaptive classifier allowed for a better BCI-control as compared to the static one.” (page 12, lines 504-505)
“This overlap in the updating parameters obtained from the two studies shows that the method we propose is not only effective but also reliable.” (page 15, lines 578-579)
- In the discussion section, the relationship of the findings with the literature can be discussed in more depth. Comparisons can be made with previous studies.
We agree with the Reviewer about the importance of discussing our results in the context of previous studies. We have significantly extended the discussion, also taking into consideration specific comments by the other Reviewers. We would also like to point out that given the absence of previous studies employing adaptive classifiers for decoding specifically imagined speech, our results have been discussed in the context of previous BCI studies based on motor imagery. On the other hand, we believe the lack of previous studies related to our main research question highlights the relevance and novelty of our work.
“We speculate that these neural differences would affect the most discriminant features for the syllables decoding, similar to the effect of the adaptive classifier in enhancing sensorimotor rhythms in motor-imagery-based BCIs [37, 63, 64].” (page 16, lines 647-650)
“Similarly, using an adaptive classifier, a previous study employing a BCI based on motor imagery demonstrated an improvement in performance over 3 days [57].” (page 17, lines 654-656)
“We hypothesized that this difference might be due to the chosen UC values not being suitable for certain participants: previous studies have suggested that suboptimal up-dating parameters could hinder the co-adaptation between the user and the system [33, 34, 60, 61].” (page 17, lines 678-681)
“A possible speculation is that participants who do not benefit from the UC chosen at the group level might present less neural activation during the imagery due to a lower ability to self-regulate brain patterns. In this direction, the lack of skills to effectively modulate the sensorimotor rhythms has been associated with reduced improvements despite the use of an adaptive classifier in previous motor-imagery-based BCI studies [56, 63]. Further work will investigate whether such neural differences associated with the ability to exploit the full potential of an adaptive classifier exist also for speech imagery. ” (page 17, lines 683-690)
- Limitations of the study and suggestions for future research could be stated more specifically.
We thank the reviewer for this comment. We have added in the revised version of the manuscript the following text concerning possible future directions:
“It remains to be addressed to what extent learning dynamics and the associated neural changes elicited by training across multiple sessions differ when using a static or an adaptive classifier. ” (page 17, lines 659-661)
“Future directions will also explore the use of other classifiers suitable for adaptive decoding that might offer performance superior to the LDA classifier. Adaptive Support Vector Machine (SVM) is one candidate, shown to outperform LDA for motor imagery decoding [72]. In the context of imagined speech, static SVMs have been previously employed for offline simulations [27, 73] and in one of the few online studies [18]. Last, it is worth noting that semi-supervised or unsupervised classifiers might be more effective in shortening training time [74], increasing the ecological validity of our BCIs, and boosting co-adaptation [36, 56].” (page 18, lines 709-717)
In addition, the original version of the manuscript already included several ideas for future research in the discussion, such as:
“Another potential optimization of the adaptive classifier consists of varying the UC values over the course of the experiment, instead of being kept constant as in our study. [...]” (page 17, lines 699-700)
“This approach could constitute a further improvement of our decoder, together with integrating a feature selection step that would focus the decoding on specific regions (left central and temporal bilateral) and frequency bands (alpha to low-gamma), shown to be the most discriminant with surface EEG for syllables imagery [30].” (page 18, lines 706-709)
And a limitation of the use of individualized choice of UC:
“Although promising, it should be noted that simulations to individualize UC values require substantial amounts of data and computational resources, thus their effectiveness remains to be established in real-time control settings”
- Visual elements (figures, tables) can be made more descriptive and understandable.
We concur with the suggestion that the figures can be improved. For Figure 1, we have incorporated additional details about the duration of each step in the figure itself and in the legend as follows:
Figure 1: Experimental paradigm (Studies 1 & 2). Each trial began with the display of the trial number (1s), followed by a fixation cross (2s). Participants then were presented with a written cue (2s) indicating the syllable they had to imagine (“fo” or “ghi”). Subsequently, an empty battery appeared on the screen and participants had to start imagining pronouncing the syllable for 5 seconds, during the offline training phase, or until the battery was filled in the online control phase. Participants were given 4 seconds to rest at the end of each trial while the text “Rest” appeared on the screen. (page 6, lines 234-242)
In Figures 3 and 6, we have modified the design by using squares instead of dots in the highlighted locations, which helps visualizing the underlying values. As for Figure 4 and 5, we believe box-and-whisker plots, as well as scatter plots, provide a comprehensive representation of our data and results.
Of note, we realized that in the original version of the manuscript, Figure 4b represented values relative to the Predicted Accuracy rather than the BCI-control performance as reported in the legend and main text. The revised version of the figure includes the correct variable, whereas the corresponding statistical values reported in the text remain unchanged as they already referred to the BCI-control performance.
We would like to point out that our study does not include any table.
- Writing equations in a more orderly and aligned manner will increase readability.
We thank the Reviewer for pointing out that equations were not centered. We have now aligned the equations based on the journal template.
- The way references are cited in the text should be corrected. Citation should be given in accordance with the journal template.
We thank the reviewer for catching this inconsistency. We have carefully checked and corrected the style of all references.
- Some sentences may be complex or too technical. Try to use simpler and more understandable language.
We agree the text should be accessible to the reader and we have done our best to improve it. Accordingly, we have provided more detail about the mathematical methods used in the revised version of the manuscript (page 7). However, we would like to point out that the present study relies considerably on technical matters: the field of Brain-computer interface pertains to the neural engineering domain and we tackle the scientific question of this study with a rather technical approach. In addition, it is difficult to simplify the text without knowing exactly which sentences the Reviewer is referring to. Doing so bears the risk of undermining the level of detail required to understand our procedure and the reproducibility of our work.
- Check for grammar and spelling errors.
We regret the presence of grammar errors in the original manuscript, we have now carefully revised the text and corrected it with the help of a native English speaker.
Reviewer 2 Report
Comments and Suggestions for Authors
1. We recommend repositioning key words appropriately to ensure their arrangement reflects the conceptual relevance of the innovative aspects discussed in the manuscript.
2. We suggest uniformity in the format and size of text within images throughout the manuscript to ensure both the standardization and aesthetic appeal of the article.
3. It is advised to delve more deeply into the discussion, providing a comprehensive explanation for the superior performance of the adaptive classifier in real-time BCI experiments. Consider exploring potential neural mechanisms underlying this improvement and establishing connections with prior research findings.
4. Further exploration is recommended in the discussion to elucidate why certain participants did not exhibit improvement in BCI control when utilizing the adaptive classifier. Consider investigating individual differences, such as specific patterns in neural activity, which may contribute to this variability.
5. Emphasize in the conclusion the potential practical applications of utilizing adaptive classifiers. Explore the anticipated real-world effects on brain-computer interface systems and training methodologies, as well as the potential benefits for individuals in self-regulating brain patterns.
6. In the conclusion, underscore the user-friendliness, cost-effectiveness, and potential contributions of adaptive classifiers towards advancing multi-class BCI system development. This will aid readers in better comprehending the positive impact of this technology on practical applications.
Comments on the Quality of English Language
Minor editing of English language required
Author Response
- We recommend repositioning key words appropriately to ensure their arrangement reflects the conceptual relevance of the innovative aspects discussed in the manuscript
We would like to start by thanking the Reviewer for the comments. We agree with the Reviewer that the order of keywords could be improved, and we propose the following new order, together with the addition of a new keyword reflecting the updated title:
“Brain-Computer interface; Adaptive LDA classifier, Electroencephalography, Speech imagery, Syllables decoding”
- We suggest uniformity in the format and size of text within images throughout the manuscript to ensure both the standardization and aesthetic appeal of the article.
We thank the Reviewer for having pointed out differences in the text style between figures, we have now ensured that the same font and size of the text is used across all figures.
Of note, we realized that in the original version of the manuscript, Figure 4b represented values relative to the Predicted Accuracy rather than the BCI-control performance as reported in the legend and main text. The revised version of the figure includes the correct variable, whereas the corresponding statistical values reported in the text remain unchanged as they already referred to the BCI-control performance.
- It is advised to delve more deeply into the discussion, providing a comprehensive explanation for the superior performance of the adaptive classifier in real-time BCI experiments. Consider exploring potential neural mechanisms underlying this improvement and establishing connections with prior research findings.
We thank the Reviewer for raising the very relevant point of whether the use of an adaptive classifier could impact neural activity and the relationship with behavioral improvement. In this study, we were not interested in the first place in neural mechanisms underlying the use of the adaptive classifier, as the primary goal of the study was to validate the use of the adaptive classifier to improve BCI-control. Importantly, we did not find any difference in the decoding accuracy when comparing the online control phase using the adaptive and static classifier (Fig. 5b). The absence of a difference in the decodability between the two experimental sessions suggests that the use of the adaptive classifier is not sufficient to increase the neural distance between the two syllables in a single session. In other words, the neural activity associated with the use of both classifiers is comparable. Some of these considerations are already present in the original version of the manuscript (“This discrepancy suggests that the improvement observed in real-time is solely due to an optimization on the decoder side and it is not accompanied by an adaptation of the underlying neural activity when using one or the other classifier. If a difference had emerged along the course of the experiment, it would have been likely due to the neu-ral data being more discriminable (between the two classes) during the adaptive ses-sion. Ultimately, this would be reflected in a higher CV accuracy.”, page 16, lines 633-639).
Following the Reviewer's comment, we propose a possible neurophysiological interpretation as follows: “[..] boosting real-time control with an adaptive classifier has the potential of enhancing neural discriminability in the medium term. We speculate that these neural differences would affect the most discriminant features for the syllables decoding, similar to the effect of the adaptive classifier in enhancing sensorimotor rhythms in motor-imagery-based BCIs [37, 63, 64].” (page 16, lines 647-650)
“It remains to be addressed to what extent learning dynamics and the associated neural changes elicited by training across multiple sessions differ when using a static or an adaptive classifier. ” (page 17, lines 659-661)
- Further exploration is recommended in the discussion to elucidate why certain participants did not exhibit improvement in BCI control when utilizing the adaptive classifier. Consider investigating individual differences, such as specific patterns in neural activity, which may contribute to this variability.
We thank the Reviewer for this useful comment and completely agree that the differences we found in degree of improvement is an interesting and relevant result. We believe individual differences could be attributed to the use of the same UC values for all participants. Indeed by tuning the UC values at the individual level, the predicted accuracy improved uniformly in all participants. This result is already discussed in the original version of the manuscript as follows: “We then tested whether optimizing the UC values at the individual level resulted in a more uniform increase and found that individualizing the UC parameters improved PA in all participants.” (page 20, lines 681-683).
It is difficult to attribute this effect to a specific neural inter-individual difference and as mentioned in the previous comment, characterizing the neural correlates of the adaptive classifier use was outside the scope of the present paper. Nevertheless, we now provide a potential interpretation of this result based on the previous literature on motor imagery as follows:
“We hypothesized that this difference might be due to the chosen UC values not being suitable for certain participants: previous studies have suggested that suboptimal updating parameters could hinder the co-adaptation between the user and the system system [33, 34, 60, 61].” (page 17, lines 678-681)
“A possible speculation is that participants who do not benefit from the UC chosen at the group level might present less neural activation during the imagery due to a lower ability to self-regulate brain patterns. In this direction, the lack of skills to effectively modulate the sensorimotor rhythms has been associated with reduced improvements despite the use of an adaptive classifier in previous motor-imagery-based BCI studies [56, 63]. Further work will investigate whether such neural differences associated with the ability to exploit the full potential of an adaptive classifier exist also for speech imagery. ” (page 17, lines 683-690)
5. Emphasize in the conclusion the potential practical applications of utilizing adaptive classifiers. Explore the anticipated real-world effects on brain-computer interface systems and training methodologies, as well as the potential benefits for individuals in self-regulating brain patterns.
We have taken into account the Reviewer’s comment and added the following sentence in the Conclusions section “Given their superior decoding performance on the machine side and the potential implication of fostering engagement and learning to self-regulate brain patterns on the user side, adaptive classifiers offer a time-saving approach to improve BCI-controllability.” (page 18, lines 737-740)
- In the conclusion, underscore the user-friendliness, cost-effectiveness, and potential contributions of adaptive classifiers towards advancing multi-class BCI system development. This will aid readers in better comprehending the positive impact of this technology on practical applications.
We have addressed these practical aspects together with the previous comment.
In addition, we believe the original version of the manuscript already included a thorough discussion on the implications of adaptive classifiers for the development of multi-syllables BCIs as follows: “Moreover, the use of adaptive classifiers is particularly relevant for surface-EEG recordings as an effective means to increase the number of decoding classes. Most of the current BCIs are limited to binary classification due to the necessity of having enough training data to counterbalance the low signal-to-noise ratio. By using data from the real-time control as training data, an adaptive classifier shortens the training time and favors the development of multi-class BCIs [74, 80]. ” (page 18, lines 732-737)
Reviewer 3 Report
Comments and Suggestions for Authors
The main contributions of this study must be listed at the end of Section 1.
Figure 1, representing the framework, should be improved in order to be more visually descriptive.
Why the actual paradigm was chosen, and why were not included more complex and difficult paradigms in the experiments?
The introduction is long; it will be better to split it out to add a new section focused on "Related Works".
The actual section (Materials and methods) is long, and some subsubsection are quite short. This section could be substantially improved.
The citations do not use the numeric format described in the author's guidelines and recommendations.
The mathematical foundations should be explained deeper to better understand the manuscript and justify the proposed method, considering that modern Deep Learning methods outperform classical classifiers as the LDA-based algorithms.
The bibliographical references must be updated, and recent high-impact research from last year should be taken into account.
Comments on the Quality of English Language
The manuscript requires deep proofreading.
Author Response
- The main contributions of this study must be listed at the end of Section 1. (last paragraph)
We would like to start by thanking the Reviewer for the comments. We have taken into account this remark and have now added the main contribution at the end of the introduction:
“This research addresses the limited effectiveness of current approaches to decode imagined speech in real-time, an important challenge that deserves prompt solutions.” (page 3, lines 111-112)
- Figure 1, representing the framework, should be improved in order to be more visually descriptive.
We agreed with the Reviewer’s comment that Figure 1 lacked information. Accordingly, we have revised the figure by including additional details concerning the duration and content of each step of the experimental paradigm in the figure, and updated the figure’s legend as follows:
Figure 1: Experimental paradigm (Studies 1 & 2). Each trial began with the display of the trial number (1s), followed by a fixation cross (2s). Participants then were presented with a written cue (2s) indicating the syllable they had to imagine (“fo” or “ghi”). Subsequently, an empty battery appeared on the screen and participants had to start imagining pronouncing the syllable for 5 seconds, during the offline training phase, or until the battery was filled in the online control phase. Participants were given 4 seconds to rest at the end of each trial while the text “Rest” appeared on the screen. (page 6, lines 234-242)
- Why the actual paradigm was chosen, and why were not included more complex and difficult paradigms in the experiments?
We understand the Reviewer’s comment and we are pleased to provide the rationale behind our experimental choice. First, the present study builds upon previous work from our group that employed this paradigm (i.e. a binary BCI with a visual feedback consisting of a battery, Bhadra et al., BioRxiv 2023). The dataset derived from this previous study (Study 1) was employed to develop the method presented in this manuscript, imposing the use of the same paradigm for experimentally testing its effectiveness (Study 2).
Concerning the choice of the visual feedback, given the demanding cognitive task users were engaged in, we purposefully chose a visual feedback that would minimally interfere with performing the syllable imagery. In addition, we opted for a feedback that could intuitively convey insights into the user’s performance. In this sense, our feedback aligns with the use of a bar/cursor moving to the left or the right side of the screen, classically employed in motor-imagery based BCI paradigms. In our case, to avoid any potential confounds due to the differences in the visual feedback between the two classes, such as the direction of the feedback’s movement, we decided to employ the exact same feedback for both imagined syllables. We have added a sentence explaining these motivations as follows:
“The same visual feedback was used for both classes to minimize potential confounds that could arise from different sensory -afferent- information associated with each of the two syllables (e.g. a different movement direction of the feedback). The choice of the battery ensured the feedback to be provided in a simple and intuitive way, to minimize cognitive load associated with the visual stimulation and to interfere the least with the performance of the imagery.” (pages 4, lines 179-184)
- The introduction is long; it will be better to split it out to add a new section focused on "Related Works".
We acknowledge that the introduction includes several topics, imposed by the fact that the subject of our study is intrinsically multidisciplinary. The length is approximately 1000 words, which aligns with the upper limit imposed by some editors. We did our best to reduce this part by removing (lines 36-37, 44-47, 56-58 of the original manuscript) or making some sentences of the first section. Furthermore, we have taken into account the Reviewer’s comment and split the introduction in two subsections, entitled “1.1 Brain-computer interfaces for speech decoding” and “1.2 Non-stationarity of EEG signals and adaptive classifiers”.
- The actual section (Materials and methods) is long, and some subsubsections are quite short. This section could be substantially improved.
We agree that the Material and Methods section is rather long and the division into subsections is sometimes avoidable. Accordingly, we have removed the subtitle “2.2.1 Syllable imagery”, and reduced the number of subsections of 2.5 from the initial four to one.
We have tried our best to improve this section, carefully taking into account the suggestions from Reviewer 1 to provide more detailed information about the methods used, in the direction of the present comment. Because our study proposes a particularly technical approach, we believe a substantial level of details in the Materials and Methods is essential to ensure the reproducibility of our work.
- The citations do not use the numeric format described in the author's guidelines and recommendations.
We thank the Reviewer for pointing out this issue, we have now corrected the citation style according to the journal’s guidelines.
- The mathematical foundations should be explained deeper to better understand the manuscript and justify the proposed method, considering that modern Deep Learning methods outperform classical classifiers as the LDA-based algorithms.
We appreciate the Reviewer's observation regarding the potential superiority of Deep Learning approaches over the LDA classifier employed in our study. This is indeed a timely and pertinent consideration given the current approaches used to decode speech.
From a technical standpoint, LDA classifiers offer advantages such as smaller datasets required for training the model and lower computational resources, resulting in faster computation compared to deep learning methods. This optimization of computational resources is particularly relevant in the context of our relatively straightforward classification task, i.e. a binary one, and given the requirements of the adaptive classifier to be updated in real-time. Importantly, the adaptive approach we propose has, as one of the primary purposes, decreasing the training time, which speaks against the use of Deep Learning methods.
We have acknowledged Reviewer’s comment and extended the motivation of choosing the LDA in the methods part (page 5, lines 217-223) as follows:
“The choice of this classifier was motivated by several studies showing the effectiveness of the adaptive version in improving BCI performance as compared to the static version [32, 36, 46, 47, 56, 57]. In addition, a LDA classifier offers advantages such as a smaller dataset required for training the model and lower computational resources, resulting in faster computation compared to alternative methods such as deep learning. These characteristics were particularly relevant given the real-time updating of the adaptive classifier and one of the primary goals of our study being to reduce the experimental training time. This computationally parsimonious choice is also well-suited to our relatively straightforward binary classification task.” (page 5, lines 202-208)
Taking into consideration the present comment and comment 8 from Reviewer 1
Future directions will also explore the use of other classifiers suitable for adaptive decoding that might offer performance superior to the LDA classifier. Adaptive Support Vector Machine (SVM) is one candidate, shown to outperform LDA for motor imagery decoding [72]. In the context of imagined speech, static SVMs have been previously employed for offline simulations [27, 73] and in one of the few online studies [18]. Last, it is worth noting that semi-supervised or unsupervised classifiers might be more effective in shortening training time [74], increasing the ecological validity of our BCIs, and boosting co-adaptation [36, 56]. (page 18, lines 709-717)
Concerning the mathematical foundation, we added more detailed information also taking into consideration comment 2 from Reviewer 4.
“The inverse of the covariance matrix Σ(t)-1 can be computed by applying the Woodbury matrix identity [58], stating that for a given matrix B:
where A, U and V are conformable variables, ⊤ indicates the transpose operator, its inverse B^(-1) can be obtained as:
Combining equations (5) - (7), we can represent Σ(t) as the matrix B (equation 8) and identify the other terms of equation (7) as follows:
- The bibliographical references must be updated, and recent high-impact research from last year should be taken into account
We agree with the Reviewer that the most recent work related to our study must be cited. Accordingly, we report the references of high-impact scientific work in the field of speech BCI (e.g. Metzger et al. Nature 2023, Willet et al. Nature 2023). However, these studies employ invasive, rather than non-invasive, brain recordings, and decode attempted rather than imagined speech. These important methodological differences and the different scope make this previous work less relevant for the interpretation of our results.
Following the Reviewer's request we have added in the introduction the following references of recent studies related to speech imagery decoding published or deposited on preprint repositories:
- Duan, Y., Zhou, J., Wang, Z., Wang, Y.-K., Lin, C.-T.: DeWave: Discrete EEG Waves Encoding for Brain Dynamics to Text Translation, http://arxiv.org/abs/2309.14030, (2023)
- Pan, H., Li, Z., Tian, C., Wang, L., Fu, Y., Qin, X., Liu, F.: The LightGBM-based classification algorithm for Chinese characters speech imagery BCI system. Cogn Neurodyn. 17, 373–384 (2023). https://doi.org/10.1007/s11571-022-09819-w
- Antony, M.J., Sankaralingam, B.P., Mahendran, R.K., Gardezi, A.A., Shafiq, M., Choi, J.-G., Hamam, H.: Classification of EEG Using Adaptive SVM Classifier with CSP and Online Recursive Independent Component Analysis. Sensors (Basel). 22, 7596 (2022). https://doi.org/10.3390/s22197596
In addition, we added the following references:
- Sannelli, C., Vidaurre, C., Müller, K.-R., Blankertz, B.: Ensembles of adaptive spatial filters increase BCI performance: an online evaluation. J. Neural Eng. 13, 046003 (2016). https://doi.org/10.1088/1741-2560/13/4/046003
- Blankertz, B., Sannelli, C., Halder, S., Hammer, E.M., Kübler, A., Müller, K.-R., Curio, G., Dickhaus, T.: Neurophysiological predictor of SMR-based BCI performance. NeuroImage. 51, 1303–1309 (2010). https://doi.org/10.1016/j.neuroimage.2010.03.022
- Koizumi, K., Ueda, K., Nakao, M.: Development of a Cognitive Brain-Machine Interface Based on a Visual Imagery Method. In: 2018 40th Annual International Conference of the IEEE Engineering in Medicine and Biology Society (EMBC). pp. 1062–1065 (2018)
Some references that appeared in the original version of the manuscript do not appear in the revised version as they pertained to parts of the introduction that have been removed (see comment #4).
Additional Remark: Of note, we realized that in the original version of the manuscript, Figure 4b represented values relative to the Predicted Accuracy rather than the BCI-control performance as reported in the legend and main text. The revised version of the figure includes the correct variable, whereas the corresponding statistical values reported in the text remain unchanged as they already referred to the BCI-control performance.
Reviewer 4 Report
Comments and Suggestions for Authors
1. Before conducting LDA classification on EEG signals, were there any preprocessing steps applied? This would likely involve addressing noise that could potentially impact the discriminative performance.
2. Some parameters in the text should be consistently represented with the parameters in the functions. For instance, 'UCμ' is presented in regular font in the text, while inside the function, 'UCμ' is in italics. Additionally, there is no explanation provided for certain parameters in the function.
3. The image in Figure 3.(d) appears distorted, and some parts of the figure are unclear. The intended meaning is somewhat challenging to comprehend. Could the representation method be revised for clarity? Similar issues are also observed in Figures 4 and 5.
4. The meaning represented by the red and black dots in Figure 6.(a)-(c) pertains to the selected UC-pair. Would it be more appropriate to use red and black borders instead of dots?
5.A comparison with past literature should be incorporated into the results.
Author Response
- Before conducting LDA classification on EEG signals, were there any preprocessing steps applied? This would likely involve addressing noise that could potentially impact the discriminative performance.
We would like to start by thanking the Reviewer for the comments. We thank the Reviewer for pointing out that pre-processing steps were missing. We have added the information in this regard at page 5, lines 227-228 as follows:
“Prior to feature extraction, a 50 Hz notch filter and Common Average Reference (CAR) were applied to the data.” (page 5, lines 212-213)
- Some parameters in the text should be consistently represented with the parameters in the functions. For instance, 'UCμ' is presented in regular font in the text, while inside the function, 'UCμ' is in italics. Additionally, there is no explanation provided for certain parameters in the function.
We thank the Reviewer for these remarks. We now report all equations’ parameters in italics in the text. Regarding the explanation for these parameters, we acknowledge more details should be provided. Accordingly we have expanded it as follows:
“The inverse of the covariance matrix Σ(t)-1 can be computed by applying the Woodbury matrix identity [58], stating that for a given matrix B:
where A, U and V are conformable variables, ⊤ indicates the transpose operator, its inverse B^(-1) can be obtained as:
Combining equations (5) - (7), we can represent Σ(t) as the matrix B (equation 8) and identify the other terms of equation (7) as follows:
- The image in Figure 3.(d) appears distorted, and some parts of the figure are unclear. The intended meaning is somewhat challenging to comprehend. Could the representation method be revised for clarity? Similar issues are also observed in Figures 4 and 5.
The figures mentioned by the Reviewer are box-and-whisker plots, a standard and thorough method to graphically represent the data distribution. We have added more details about the information conveyed by this kind of representation within the legend of the first figure that employ a box-and-whisker plot (Figure 3d):
“Boxes represent the interquartile range (IQR), with the horizontal line indicating the median, and whiskers extending to data points that are within 1.5 x the IQR from the upper and lower quartile. Individual points represent data from a single participant, and gray lines connect data points from the same participant.”
Concerning the distortion of the image mentioned by the Reviewer, we regret not being able to identify the problem, maybe this is due to a technical issue on the file visualization ?
In addition, we have also optimized figures 3,4 and 5 based on Reviewer 2 comments, ensuring the text format is the same across all figures.
Of note, we realized that in the original version of the manuscript, Figure 4b represented values relative to the Predicted Accuracy rather than the BCI-control performance as reported in the legend and main text. The revised version of the figure includes the correct variable, whereas the corresponding statistical values reported in the text remain unchanged as they already referred to the BCI-control performance.
- The meaning represented by the red and black dots in Figure 6.(a)-(c) pertains to the selected UC-pair. Would it be more appropriate to use red and black borders instead of dots?
We thank the reviewer for this useful suggestion, indeed using borders helps visualizing the underlying values. Accordingly, we have replaced dots with red and black squares in Figure 6 a-c.
5. A comparison with past literature should be incorporated into the results.
We acknowledge the importance of discussing our results in light of previous evidence, and we did our best to improve this aspect. Accordingly, we have expanded several points of the discussion, also based on the other Reviewer’s comments as follows:
“We speculate that these neural differences would affect the most discriminant features for the syllables decoding, similar to the effect of the adaptive classifier in enhancing sensorimotor rhythms in motor-imagery-based BCIs [37, 63, 64].” (page 16-17, lines 647-650).
“Similarly, using an adaptive classifier, a previous study employing a BCI based on motor imagery demonstrated an improvement in performance over 3 days [57].” (page 17, lines 654-656)
“We hypothesized that this difference might be due to the chosen UC values not being suitable for certain participants: previous studies have suggested that suboptimal up-dating parameters could hinder the co-adaptation between the user and the system [33, 34, 60, 61].” (page 17, lines 678-681)
“A possible speculation is that participants who do not benefit from the UC chosen at the group level might present less neural activation during the imagery due to a lower ability to self-regulate brain patterns. In this direction, the lack of skills to effectively modulate the sensorimotor rhythms has been associated with reduced improvements despite the use of an adaptive classifier in previous motor-imagery-based BCI studies [56, 63]. Further work will investigate whether such neural differences associated with the ability to exploit the full potential of an adaptive classifier exist also for speech imagery. ” (page 17, lines 683-690)
“Future directions will also explore the use of other classifiers suitable for adaptive decoding that might offer performance superior to the LDA classifier. Adaptive Support Vector Machine (SVM) is one candidate, shown to outperform LDA for motor imagery decoding [72]. In the context of imagined speech, static SVMs have been previously employed for offline simulations [27, 73] and in one of the few online studies [18]. Last, it is worth noting that semi-supervised or unsupervised classifiers might be more effective in shortening training time [74], increasing the ecological validity of our BCIs, and boosting co-adaptation [36, 56].” (page 18, lines 709-717)
We also would like to point out that given the lack of previous studies employing BCI based on imagined speech, we had to base the interpretation of our results on past literature on BCI in the motor domain. However, we believe this also highlights the novelty of our study.
Round 2
Reviewer 1 Report
Comments and Suggestions for Authors
As a result of the review, the comments about the study are as follows;
• In general, the requested revisions have been made.
• The headings “1.1 Brain-computer interfaces for speech decoding” and “1.1 Non-stationarity of EEG signals and adaptive classifiers” are numbered the same.
• In Figure 3, the fonts of some texts are too small. It can be published after minor changes.
Comments on the Quality of English Language
Minor editing of English language required
Author Response
We would like to thank the Reviewer for providing us these additional comments, that we have now integrated in the revised version of the manuscript. The headings have been corrected as well as we increased the font size of the axis and color bar in Figure 3 a-b-c. To ensure style consistency across the entire manuscript, we implemented the same changes in Figure 6 a-b-c.
We believe this and the previous round of comments we received has contributed to improving our manuscript.
Figure 3. Simulation analyses to optimize the parameters for the adaptive LDA classifier.
Figure 6. Simulation analyses on the real-time experiment dataset.
Reviewer 2 Report
Comments and Suggestions for Authors
The author has made in-depth revisions to the comments of the reviewers, and their efforts are commendable. I believe that this version of the manuscript meets the requirements of the journal.
Author Response
We would like to thank the Reviewer for this positive response and for having appreciated our efforts. We believe the previous round of comments we received has considerably contributed to improving our manuscript.
Reviewer 3 Report
Comments and Suggestions for Authors
The authors have addressed all my previous remarks correctly. I have no further comments.
Comments on the Quality of English Language
English is fine
Author Response
We would like to thank the Reviewer for this positive response. We believe the previous round of comments we received has considerably contributed to improving our manuscript.
Reviewer 4 Report
Comments and Suggestions for Authors
Authors revised the draft according the reviewer’s comments.
Author Response

(The authors gave the same response as above.)
